# Salivary Oxidative Stress Biomarkers in Peri-Implant Disease: A Systematic Review and Meta-Analysis

**DOI:** 10.3390/ijms262311269

**Published:** 2025-11-21

**Authors:** Paul Șerban Popa, Gabriel Valeriu Popa, Kamel Earar, Claudia Elisabeta Popa-Cazacu, Mădălina Nicoleta Matei

**Affiliations:** Department of Dental Medicine, “Dunărea de Jos” University Galați, 800008 Galați, Romania; paul.popa@ugal.ro (P.Ș.P.); kamel.earar@ugal.ro (K.E.); cc445@ugal.ro (C.E.P.-C.); madalina.matei@ugal.ro (M.N.M.)

**Keywords:** peri-implantitis, oxidative stress, salivary biomarkers, malondialdehyde, total antioxidant capacity, meta-analysis, dental implants, non-invasive diagnostics

## Abstract

Peri-implantitis, a common biological complication of dental implants, is characterized by soft-tissue inflammation and progressive bone loss. Oxidative stress is increasingly implicated in its pathogenesis, yet the diagnostic potential of salivary redox biomarkers remains unclear. This study’s objective was to assess the association between salivary malondialdehyde (MDA) and total antioxidant capacity (TAC) and peri-implant disease via a pre-registered, quantitative meta-analysis of previously published studies using predefined statistical criteria. Following a priori PROSPERO registration, we systematically searched PubMed, Scopus, and Web of Science (2004–September 2025), extracted data in duplicate, and pooled effects using random-effects models; certainty of evidence was appraised with GRADE (Grading of Recommendations Assessment, Development and Evaluation) and risk of bias with ROBINS-I (Risk Of Bias In Non-randomized Studies of Interventions)/QUADAS-2 (Quality Assessment of Diagnostic Accuracy Studies-2). Twelve studies were included qualitatively: seven (n = 726) contributed to MDA and five (n = 485) to TAC meta-analyses. Peri-implant disease was associated with elevated MDA (SMD = 1.64, 95% CI 1.39–1.88) and reduced TAC (SMD = −1.88, 95% CI −2.17 to −1.58); statistical heterogeneity was not detected, and results were robust to sensitivity and exploratory assay-based subgroup analyses. Salivary MDA and TAC show consistent, large, standardized differences in peri-implant disease; however, observational designs, assay variability, and the absence of validated diagnostic thresholds warrant cautious interpretation and prospective validation before clinical adoption.

## 1. Introduction

Dental implants are widely used for the rehabilitation of edentulous spaces, offering predictable outcomes and long-term success. However, biological complications such as peri-implant diseases—particularly peri-implant mucositis and peri-implantitis—pose ongoing challenges to implant longevity and patient outcomes. Peri-implantitis, characterized by progressive bone loss in the presence of inflammation, affects an estimated 20% of implants and up to 40% of patients over time [1]. While its multifactorial etiology is well-recognized, involving microbial biofilms, host immune response, and systemic factors, its pathophysiology remains incompletely understood [2].

Increasing attention has been paid to the role of oxidative stress in peri-implant tissue breakdown. Oxidative stress refers to an imbalance between reactive oxygen species (ROS) and the antioxidant defense system, which can result in damage to lipids, proteins, and nucleic acids. In periodontal disease, elevated ROS levels have been linked to alveolar bone loss, collagen degradation, and dysregulation of host immune responses [3,4]. By analogy, similar mechanisms are hypothesized in peri-implant disease, although the evidence base is more limited. Although microbial biofilms initiate peri-implant inflammation, oxidative stress (OS) appears to play a significant intermediary role in tissue breakdown. Host immune cells—especially polymorphonuclear neutrophils (PMNs)—respond to persistent bacterial stimuli by producing reactive oxygen species (ROS). In conditions of chronic or excessive ROS, neighboring peri-implant tissues suffer collateral damage: osteoblasts may be impaired, osteoclast activity enhanced, and extracellular matrix degraded [5,6].

Saliva offers a non-invasive diagnostic medium capable of reflecting both local and systemic redox status. Salivary biomarkers such as malondialdehyde (MDA)—a marker of lipid peroxidation—and total antioxidant capacity (TAC)—a measure of overall antioxidant defense—have shown promise in periodontal disease monitoring and are increasingly being studied in peri-implant conditions [7,8,9]. Additional oxidative stress biomarkers, including 8-hydroxy-2′-deoxyguanosine (8-OHdG) and superoxide dismutase (SOD), have been examined for diagnostic relevance in peri-implant disease but were not quantitatively synthesized here [10,11,12,13,14].

While previous systematic reviews have evaluated oxidative stress biomarkers in peri-implant crevicular fluid, none to date have quantitatively synthesized findings from salivary biomarker studies. Given saliva’s accessibility, clinical practicality and patient acceptability, this represents an important gap in the literature. Furthermore, the lack of consensus over both threshold values and assay types to biomarker validity, cut-off values, and inter-study comparability underscores the need for a rigorous evidence synthesis.

The aim of this systematic review and meta-analysis is to evaluate the association between salivary oxidative-stress biomarkers—specifically MDA and TAC—and peri-implant disease by applying predefined statistical criteria in a quantitative synthesis of previously published studies. Secondary objectives include identifying methodological sources of heterogeneity, assessing the certainty of evidence using GRADE criteria, and exploring the potential for salivary redox markers to serve as diagnostic adjuncts in implant dentistry.

A schematic illustration of oxidative stress mechanisms implicated in peri-implant tissue degradation is shown in Figure 1. The interplay between bacterial challenge, polymorphonuclear neutrophil activation, and redox imbalance (via ROS, RNS <reactive nitrogen species>, MDA elevation and TAC depletion) highlights the biological plausibility of salivary oxidative stress biomarkers as candidate diagnostic indicators.

## 2. Materials and Methods

### 2.1. Protocol and Registration

This systematic review and meta-analysis was conducted following the Preferred Reporting Items for Systematic Reviews and Meta-Analyses (PRISMA) 2020 guidelines [18]. The review protocol was developed a priori and registered in PROSPERO (CRD420251117832) [https://www.crd.york.ac.uk/PROSPERO/view/CRD420251117832] (accessed on 3 September 2025). To ensure full transparency, the original PROSPERO submission form has been included as Appendix A. All methodological steps were pre-defined and aligned with established systematic review standards to ensure transparency and reproducibility. Surrogate biomarker associations were analyzed instead of the originally intended clinical endpoints of implant survival or failure. No eligible longitudinal studies reporting implant outcomes or predictive performance were identified. This represents a deviation from the protocol and is documented in Appendix A.

### 2.2. Focused Question and Objectives

The primary objective was to assess whether salivary oxidative stress biomarkers are associated with the presence of peri-implant disease. Due to the absence of eligible longitudinal studies, the review did not evaluate predictive performance for implant survival or disease progression. This review was structured according to the PICOS model:Population (P): Adult patients with dental implants, either healthy or diagnosed with peri-implantitis or peri-implant mucositis.Intervention (I): Quantitative measurement of salivary oxidative stress biomarkers (e.g., malondialdehyde [MDA], total antioxidant capacity [TAC], 8-hydroxydeoxyguanosine [8-OHdG]), defined as values reported as absolute concentration, normalized units (e.g., per protein or volume), or assay-specific readouts (e.g., relative fluorescence). Saliva was selected as a non-invasive matrix supported by prior diagnostic overviews [7,8]. For the purposes of this review, “peri-implant disease” was defined as either peri-implant mucositis (reversible inflammatory changes in the peri-implant mucosa without radiographic bone loss) or peri-implantitis (inflammatory changes with concomitant radiographic evidence of supporting bone loss), according to each study’s definition. Studies including either condition, or both, were eligible for inclusion. When a study reported both conditions as separate groups, we extracted data for each comparison (vs healthy controls) if reported in sufficient detail; otherwise, the study’s overall diseased group was used. For meta-analytic pooling, peri-implant mucositis and peri-implantitis groups were combined under the umbrella term “peri-implant disease.” Diagnostic definitions followed the 2017 World Workshop on the Classification of Periodontal and Peri-Implant Diseases and Condition [1]. The potential implications of combining these conditions are addressed in the Discussion.Comparison (C): Healthy controls with no signs of peri-implant disease.Outcome (O): Association between biomarker levels and disease presence or implant success/failure.Study design (S): Observational studies (cross-sectional, case–control, cohort), and experimental clinical studies.

### 2.3. Information Sources and Search Strategy

A comprehensive electronic search was conducted in the following databases: PubMed, Scopus, and Web of Science. The final search was performed in September 2025. The search strategy combined MeSH terms and free-text keywords using Boolean operators:Search terms: (“saliva” OR “salivary”) AND (“oxidative stress” OR “oxidative damage” OR “redox status”) AND (“biomarker” OR “marker” OR “indicator”) AND (“malondialdehyde” OR “MDA” OR “total antioxidant capacity” OR “TAC” OR “8-hydroxydeoxyguanosine” OR “8-OHdG”) AND (“peri-implantitis” OR “peri-implant mucositis” OR “dental implants”)

The search was limited to the following criteria:Language: English onlyPublication period: January 2004 to September 2025Study type: Original research articles (excluding reviews, case reports, conference abstracts, in vitro and animal studies)

Gray literature, dissertations, and clinical trial registries were not searched. Reference lists of all included full-text studies and relevant reviews were screened manually for additional eligible studies. The complete search strings and syntax for each database are provided in Appendix A. Although the protocol prespecified searching PubMed/MEDLINE, Embase, Web of Science Core Collection, Scopus, and CENTRAL, the final search omitted direct Embase and CENTRAL queries due to anticipated high duplication with Scopus and PubMed; preliminary scoping confirmed no unique eligible records from these sources. Gray literature and trial registry searches were also not performed, as preliminary screening suggested minimal relevance for salivary biomarker quantification studies. Both deviations are documented in Appendix A.

### 2.4. Eligibility Criteria

Studies were eligible for inclusion if they met all the following criteria:Human studies including participants with dental implantsSalivary biomarkers related to oxidative stress were quantified (MDA, TAC, 8-OHdG, or related)Comparison between healthy and diseased implant conditionsQuantitative outcomes reported or extractable (means, SDs, or AUC)Original full-text research articles (excluding reviews, case reports, conference abstracts, in vitro and animal studies).

Eligibility criteria were pre-specified consistent with PRISMA and Cochrane guidance [18,19,20].

### 2.5. Study Selection and Data Extraction

After removal of duplicates using Zotero (version 6.0), two reviewers independently screened titles and abstracts. Full texts of potentially relevant studies were retrieved and assessed for eligibility. Discrepancies were resolved by consensus or adjudicated by a third reviewer when necessary. A standardized data extraction form was used to collect: authors, year, study design, population characteristics, number of implants or patients, biomarker(s) evaluated, method of analysis (e.g., ELISA (enzyme-linked immunosorbent assay), spectrophotometry), and primary outcomes. When numerical data were incomplete, corresponding authors were contacted. If no response was received, data were estimated from figures when possible or excluded from quantitative synthesis with justification. When studies reported both patient-level and implant-level data, patient-level estimates were prioritized. If multiple articles reported data from the same cohort, the most comprehensive or recent dataset was included. When numerical data were not available in tabular form, digital extraction from published figures was performed using WebPlotDigitizer version 5.2 (Ankit Rohatgi). These approaches were unsuccessful for one study [14], which was therefore included only in the qualitative synthesis. In cases where estimates could not be reliably recovered, studies were excluded from quantitative synthesis but retained for qualitative review

Only studies reporting quantitative outcomes (means, standard deviations, or values convertible to SMD, standardized mean difference) for both disease and control groups were included in the meta-analysis. Studies with insufficient or non-convertible data were retained for qualitative synthesis only. Exploratory subgroup analyses were conducted post hoc to examine whether analytical method (e.g., ELISAELISA vs. TBARS, thiobarbituric acid reactive substances or colorimetric assays) influenced the magnitude of effect estimates. Sensitivity analyses were performed by sequentially excluding each study to test the robustness of pooled estimates. MDA units varied across studies, including μmol/L, nmol/mL, and nmol/mg protein, depending on the assay method. This limits direct interpretability and increases heterogeneity risk even if SMD was used. While SMD was used to account for differences in scale, no formal unit conversion or normalization was performed prior to meta-analysis. This may introduce additional heterogeneity not captured by I^2^, and subgroup analysis by assay type was used as a partial proxy to address this concern.

### 2.6. Risk-of-Bias Assessment 

The ROBINS-I tool [21] was applied to assess risk of bias in observational studies across seven domains: confounding, participant selection, classification of interventions, deviations from intended interventions, missing data, outcome measurement, and selection of reported results. For diagnostic-style studies, the QUADAS-2 tool [22] was used, evaluating four domains: patient selection, index test, reference standard, and flow/timing. Risk of bias was assessed at the study level using design-appropriate tools. ROBINS-I was applied to all observational studies and non-randomized interventional studies (n = 7), assessing bias across seven domains with judgments of Low, Moderate, Serious, or Critical risk. QUADAS-2 was applied to studies evaluating salivary biomarkers as diagnostic tests distinguishing diseased from healthy participants (n = 5), covering four domains (Patient Selection, Index Test, Reference Standard, Flow/Timing). One study [23] included both a non-randomized intervention and a diagnostic accuracy component and was therefore assessed with both ROBINS-I and QUADAS-2. No randomized controlled trials were identified, so the Cochrane RoB 2 tool was not applicable. 

Certainty ratings started at Low for observational studies and could be downgraded for study limitations, inconsistency, indirectness, imprecision, or publication bias, and upgraded for large effect size, dose–response, or if all plausible confounding would reduce a demonstrated effect.

Detailed domain-level judgments and signaling questions are included in Appendix A.

### 2.7. Data Synthesis and Statistical Analysis 

A meta-analysis was performed separately for MDA and TAC. Standardized mean differences (SMDs) with 95% confidence intervals (CI) were calculated using a random-effects model due to expected clinical and methodological heterogeneity [19]. Heterogeneity was quantified using the I^2^ statistic and interpreted as low (I^2^ < 30%), moderate (30–50%), or substantial (I^2^ > 50%) [20]. Publication bias was assessed using Egger’s regression test [24]. Due to the small number of included studies per outcome (<10), Egger’s regression test was interpreted cautiously. Funnel plot asymmetry was evaluated visually, and publication bias was considered undetected unless corroborated by both visual and statistical evidence. Sensitivity analyses were conducted by sequentially omitting each study. Subgroup analyses were performed based on an assay method (e.g., ELISA vs. spectrophotometry). A random-effects model (DerSimonian–Laird method) was selected a priori to account for expected methodological and clinical heterogeneity, even in the presence of low statistical heterogeneity. Where necessary, medians and interquartile ranges were converted to means and standard deviations using established statistical approximations [25]. A priori, we considered analytical method (e.g., TBARS vs. ELISA) as a potential source of heterogeneity and planned subgroup/sensitivity comparisons accordingly.

All statistical analyses were conducted using Review Manager (RevMan) version 5.4 (The Cochrane Collaboration) and MetaXL version 5.3. No user-modified scripts were used; default program settings were applied for pooled estimates, forest plots, and funnel plots. While all analyses were conducted using standard GUI-based workflows in RevMan and MetaXL, we acknowledge that these environments do not produce user-readable code. Therefore, reproducibility is limited to the graphical interface. To enhance transparency, the full input dataset, extracted data sheet, and effect estimates are available upon request. Future replications using R or Stata are planned for follow-up sensitivity analyses. A *p*-value < 0.05 was considered statistically significant. SMDs were interpreted as small (0.2), moderate (0.5), or large (≥0.8) effects [19]. A GRADE approach was applied to assess the overall certainty of evidence, considering risk of bias, inconsistency, indirectness, imprecision, and publication bias [26]. Appendix A contains the GRADE Evidence Profile Tables.

## 3. Results

### 3.1. Study Selection

A total of 83 records were identified through systematic database searches (PubMed, Scopus, and Web of Science). After the removal of 45 duplicates, 38 records remained for title and abstract screening. Of these, 35 full-text articles were retrieved and assessed in detail for eligibility. Following full-text review, 12 studies met all inclusion criteria and were included in the qualitative synthesis. Among them, seven studies were eligible for quantitative synthesis (meta-analysis) of malondialdehyde (MDA) levels [5,13,23,27,28,29,30], and five studies for total antioxidant capacity (TAC) levels [13,23,27,29,31]. One included study met all eligibility criteria and reported no statistically significant differences in TAC or MDA levels between implant and control groups; however, quantitative data (means and standard deviations) were not reported, precluding inclusion in the meta-analysis [14]. The full study selection process is detailed in the PRISMA flow diagram (Figure 2), in accordance with PRISMA 2020 guidelines.

### 3.2. Study Characteristics

The twelve included studies were published between 2013 and 2024 and collectively involved 1042 patients, with individual sample sizes ranging from 32 to 210 participants. The majority employed cross-sectional designs; three studies were interventional in design but were not randomized controlled trials; all were pre–post or controlled before–after designs and were therefore evaluated with ROBINS-I rather than RoB 2, and two were structured as diagnostic accuracy studies incorporating receiver operating characteristic (ROC) analysis. All studies assessed salivary oxidative stress biomarkers in the context of peri-implant mucositis or peri-implantitis, compared to healthy controls. No study provided sufficient longitudinal data to assess progression from mucositis to peri-implantitis. In the quantitative synthesis, peri-implant mucositis and peri-implantitis were pooled as “peri-implant disease,” due to the limited number of studies available for separate analyses. The primary biomarkers evaluated were MDA, TAC, and 8-hydroxydeoxyguanosine (8-OHdG). Measurement techniques included enzyme-linked immunosorbent assay (ELISA), colorimetric spectrophotometry, and the thiobarbituric acid reactive substances (TBARS) assay. Implant duration varied substantially across studies—from 3 months to over 12 years—reflecting both early and long-term disease progression scenarios.

### 3.3. Risk-of-Bias Assessment

ROBINS-I assessment indicated that all seven observational/non-randomized interventional studies were at overall Moderate risk of bias, most commonly due to potential confounding, selection bias, and possible lack of blinding of laboratory personnel to participants’ disease status during biomarker analysis. QUADAS-2 assessments found that all three diagnostic-design studies were at low risk of bias in the Patient Selection, Reference Standard, and Flow/Timing domains; however, the Index Test domain was judged as “Unclear” in all cases because biomarker threshold criteria were not pre-specified. Overall, these studies were considered at low concern for bias, but the lack of prespecified index test thresholds should be noted when interpreting the findings. A detailed risk-of-bias summary is presented in Appendix A.

### 3.4. Meta-Analysis of MDA Levels

Seven studies contributed to the meta-analysis of salivary MDA concentrations. The total sample size was 726 participants, including 412 patients with peri-implant disease and 314 healthy controls. Using a random-effects model, the pooled standardized mean difference (SMD) was 1.64 (95% CI: 1.39 to 1.88), indicating significantly higher MDA levels in diseased individuals (Figure 3).

All studies reported higher MDA concentrations in patients with peri-implant disease, with individual SMDs ranging from 1.2 to 2.1. No contradictory or null associations were identified. No statistical heterogeneity was observed (I^2^ = 0%), and the confidence interval was relatively narrow, suggesting consistent directionality of the association. While clinical and methodological diversity may still be present, the effect estimates were remarkably homogeneous. However, given the small sample sizes in individual studies, overall precision remains limited in terms of clinical inference. A *p*-value of <0.05 was considered statistically significant for all inferential tests. Egger’s regression (*p* = 0.172) and visual inspection of the funnel plot (Figure 4) did not suggest publication bias. Although no visual evidence of publication bias was detected, it is important to note that the statistical power of Egger’s test is limited when fewer than 10 studies are included in the meta-analysis. As such, the absence of asymmetry should be interpreted with caution, and potential small-study effects cannot be definitively ruled out.

Sensitivity analysis, performed using a leave-one-out approach, showed no significant shifts in pooled estimates, confirming the internal consistency of the findings across the included studies, though overall certainty remains constrained (Figure 5).

An exploratory subgroup descriptive analysis based on biomarker quantification method (ELISA vs. TBARS) did not demonstrate substantial variation in effect size, supporting the robustness of results across analytical platforms (Figure 6). The observed SMD exceeds the conventional threshold for a large effect size (≥0.8), underscoring a meaningful clinical association. Across assay subgroups (TBARS vs. ELISA), pooled effects remained directionally consistent and similar in magnitude, indicating that the association between higher salivary MDA and peri-implant disease was robust to the measurement platform.

### 3.5. Meta-Analysis of TAC Levels

Five studies, comprising 485 patients (281 with peri-implant disease and 204 healthy controls), were eligible for the meta-analysis of total antioxidant capacity (TAC)**.** The pooled estimate showed a significant reduction in TAC levels among diseased individuals, with a SMD of −1.88 (95% CI: −2.17 to −1.58) (Figure 7).

Individual study SMDs ranged from −1.5 to −2.4, all in the same direction. No statistical heterogeneity was observed (I^2^ = 0%), and the relatively narrow confidence intervals indicate a consistent direction of association across studies. Despite possible underlying clinical or methodological differences, the effect sizes appeared strikingly uniform. However, the limited sample sizes in individual studies constrain the precision of the pooled estimate, warranting caution when drawing clinical inferences (Figure 8). TAC values were consistently reduced in disease groups across all included studies and analytic methods. Exploratory comparisons by analytical method did not suggest material differences in the observed reduction in TAC among disease groups.

### 3.6. Certainty of Evidence (GRADE Assessment)

The GRADE evaluation indicated a low certainty of evidence for the MDA outcome, downgraded for risk of bias and imprecision. Although the pooled effect size for MDA was large (SMD = 1.64) and its 95% confidence interval was narrow (1.39–1.88), imprecision was still considered serious due to the limited total sample size (n = 726) and the small individual study sizes—most of which included fewer than 100 participants per group. According to GRADE guidance, the optimal information size (OIS) was not met, and the absence of predefined clinical decision thresholds for salivary MDA levels further contributes to uncertainty about the precision of effect estimates in guiding practice. Therefore, the imprecision domain was downgraded despite statistical significance and consistent directionality across studies. While the statistical estimate for MDA was consistent and precise, the GRADE rating remained low due to the limited total information size and methodological concerns, consistent with GRADE guidelines. The evidence for TAC was rated as moderate, downgraded for inconsistency. Despite these limitations, the directionality of findings was consistent across all studies, supporting a biologically plausible association with potential clinical relevance. Both outcomes showed large effect sizes and no serious statistical inconsistency (I^2^ = 0%). TAC was upgraded by one level for large magnitude of effect, but both remained limited by the observational design of included studies and potential residual confounding. Domain-level GRADE assessments are provided in Appendix A.

### 3.7. Qualitative Synthesis of 8-OHdG

Three studies assessed salivary 8-OHdG levels as a marker of oxidative DNA damage. All reported elevated concentrations in the peri-implant disease group. However, substantial heterogeneity in reporting formats (e.g., ng/mL vs. relative fluorescence units), sampling protocols, and absence of standard deviations prevented meta-analysis. The converging trend across studies indicates that 8-OHdG warrants further investigation as a candidate biomarker for diagnostic or prognostic use in peri-implant pathology. Standardization of assay methodology and data reporting is necessary to enable future quantitative synthesis.

### 3.8. Summary of Findings

In summary, both elevated MDA and reduced TAC levels were associated with peri-implant disease, with no detected statistical heterogeneity or small-study effects. Sensitivity and subgroup analyses confirmed the consistency and stability of the findings across methodological subgroups and patient populations. While the evidence for 8-OHdG remains preliminary, it highlights a promising biomarker pathway for future salivary diagnostics in peri-implant conditions.

## 4. Discussion

This systematic review and meta-analysis suggests that salivary oxidative stress markers—specifically elevated malondialdehyde (MDA) and reduced total antioxidant capacity (TAC)—are consistently associated with peri-implant disease. Across the included studies, patients diagnosed with peri-implantitis exhibited significantly higher MDA levels and lower TAC values compared to healthy implant controls. The pooled effect size for MDA was particularly large (SMD = 1.64), with narrow confidence intervals and no statistical heterogeneity, suggesting a consistent and perhaps clinically meaningful elevation in oxidative stress among peri-implant disease patients. While the certainty of this evidence remains low due to study design limitations and sample size considerations, the magnitude and consistency of the association warrant further exploration in prospective studies. These findings were consistent across multiple sensitivity and subgroup analyses and remained correlated despite methodological variability. Importantly, no strong evidence of publication bias was observed in funnel plots or Egger’s test, and the overall certainty of evidence was rated low to moderate based on GRADE criteria. As noted in the Results, one large observational study found no significant differences in salivary TAC or MDA between diseased and healthy implants but could not be meta-analyzed due to incomplete data. While its exclusion from the quantitative synthesis may contribute to overestimation of pooled effects, its qualitative inclusion helps contextualize the overall evidence base.

Mechanistic and narrative reviews previously suggested oxidative imbalance in peri-implant tissues, but lacked quantitative evidence [32]. A recent systematic review and meta-analysis by Wang et al. focused on oxidative biomarkers in peri-implant sulcular fluid, and reported significantly decreased glutathione peroxidase (GSH-Px), elevated myeloperoxidase (MPO), and increased MDA levels in peri-implantitis, with standardized mean differences ranging from –1.40 for GSH-Px to +0.46 for MPO and +0.28 for MDA [17]. Our results extend that evidence to salivary markers, which may provide a more accessible and non-invasive diagnostic matrix.

A recent clinical cross-sectional study by Özkan Karasu et al. evaluated salivary 8-OHdG, MDA, SOD, and GPx in peri-implantitis, peri-mucositis, and healthy groups—reporting significantly higher MDA, 8-OHdG, and SOD, with lower GPx in peri-implantitis [2]. In contrast, a large observational study found no significant salivary differences in TAC or MDA between implant and control groups, suggesting variability by assay and patient characteristics [14].

Elevated MDA reflects lipid peroxidation and membrane damage, likely driven by local inflammation and ROS generation in peri-implant tissues. TAC measures cumulative antioxidant capacity, encompassing enzymatic (e.g., SOD, GPx) and non-enzymatic systems (e.g., uric acid, glutathione) [15]. Reduced TAC may signify depleted defense mechanisms, with consequences for collagen degradation, tissue inflammation, and osteoclastic bone loss—processes observed in periodontal analog models [16]. Importantly, systemic factors like diabetes, smoking, and nutrition may confound salivary redox profiles, and although most studies adjusted for some variables, residual confounding remains possible.

This review is characterized by a comprehensive, pre-registered protocol, dual independent screening and extraction, use of validated bias tools (ROBINS-I, QUADAS-2), and standardized GRADE assessment. Subgroup analysis by collection/normalization method and sensitivity checks enhanced robustness. Nonetheless, limitations include the following:

Exclusively observational designs, limiting causal inference

Heterogeneous protocols for saliva collection and biomarker normalizationModerate risk of bias in several studiesPotential language/publication bias, despite no funnel-plot asymmetry.

A key limitation of the MDA meta-analysis is the lack of unit standardization across studies. While SMD is designed to accommodate differences in measurement scale, mixing units such as μmol/L and nmol/mg protein may still compromise comparability. This could influence the magnitude and interpretation of the pooled estimate, and future studies should report MDA values in standardized formats to allow for harmonized synthesis. Methodological variability, particularly TBARS (a composite thiobarbituric-acid reactive signal) versus ELISA-based quantification—can shift absolute concentrations, which is why we synthesized standardized mean differences. This approach supports inference on direction and relative magnitude, but it limits translation to immediate clinical cut-offs; further work must standardize pre-analytics, assay platforms, and reporting units. 

A broader narrative review noted that oxidative stress markers’ clinical utility in periodontal research remains limited due to assay variability and poor reproducibility [15].

This review could not include prospective or longitudinal studies assessing implant survival, failure, or disease progression. As such, the findings reflect cross-sectional associations between salivary oxidative stress biomarker levels and current peri-implant disease status, rather than predictive validity for future implant outcomes. The lack of prospective evidence precludes conclusions about causality or temporal relationships between biomarker levels and disease onset. The magnitude of the pooled standardized mean differences (SMD ≈ 1.6 for MDA; SMD ≈ 1.9 for TAC) indicates very large, standardized effects, corresponding to substantial separation between the biomarker distributions in peri-implant disease and healthy control groups. In conventional terms, an SMD > 0.8 is often interpreted as a large effect; here, values > 1.5 imply minimal overlap between groups. However, direct translation into absolute concentration differences or clinically actionable cut-offs is not currently feasible, as the included studies reported biomarker concentrations in heterogeneous units (e.g., μmol/L, nmol/mg protein) and used differing analytical methods. This uncertainty reinforces the need for methodological standardization and for prospective studies aimed at establishing validated diagnostic thresholds for these biomarkers. 

Although most included studies were judged at low or moderate risk of bias, the observational and non-randomized interventional designs are inherently more susceptible to confounding than randomized trials. For diagnostic studies, the absence of prespecified biomarker thresholds contributed to “Unclear” risk judgments in the Index Test domain, which may limit reproducibility and external validity.

Collectively, elevated MDA and reduced TAC support a measurable salivary redox imbalance in peri-implant disease, consistent across assays and small-study sensitivity analyses. While these findings may ultimately support non-invasive risk assessment, clinical implementation awaits standardized collection/assays and validated diagnostic thresholds in longitudinal cohorts.

This systematic review combined peri-implant mucositis and peri-implantitis under a single category (“peri-implant disease”) for meta-analysis. While this approach increased the number of studies contributing to each biomarker estimate, it introduces potential clinical heterogeneity, as peri-mucositis is a reversible inflammatory condition, whereas peri-implantitis involves irreversible bone loss. Data were too sparse to perform a robust subgroup meta-analysis; however, qualitative inspection of effect sizes suggested that the direction and magnitude of biomarker differences were generally consistent across both conditions. Future studies should report and analyze peri-implant disease stages separately to clarify whether biomarker levels differ meaningfully by severity.

A schematic summary of these proposed mechanisms and potential diagnostic applications is shown in Figure 9, illustrating how salivary MDA and TAC could complement each other as oxidative stress indicators in peri-implant disease.

## 5. Conclusions

Across 12 observational studies, peri-implant disease was associated with higher salivary MDA and lower TAC (MDA: SMD 1.64; TAC: SMD −1.88; both I^2^ = 0%). Certainty of evidence was low (MDA) to moderate (TAC). These findings support a biologically plausible salivary redox signal but, given observational designs and assay heterogeneity, they should not be used clinically without validated thresholds and prospective confirmation. Standardized protocols and longitudinal studies are the immediate priorities.

## Figures and Tables

**Figure 1 ijms-26-11269-f001:**
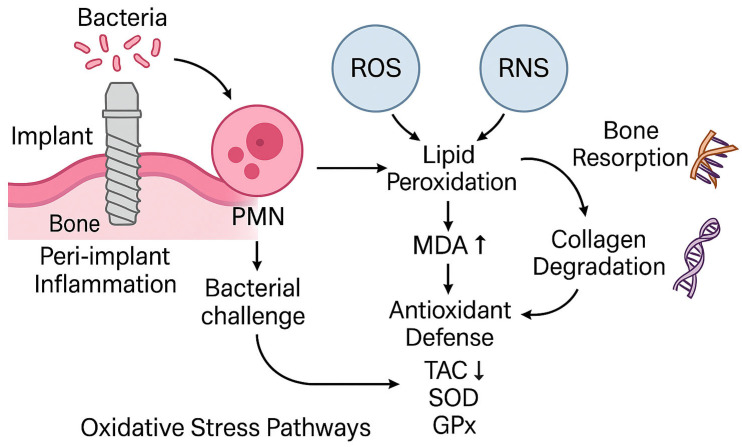
Schematic of oxidative stress pathways implicated in peri-implant tissue degradation (bacterial challenge → neutrophil activation → ROS/RNS → lipid peroxidation [MDA] increase and antioxidant depletion [TAC]). Concept informed by periodontal/peri-implant OS literature [15,16] and peri-implant OS syntheses [6,17]; diagnostic saliva context in [7,8].

**Figure 2 ijms-26-11269-f002:**
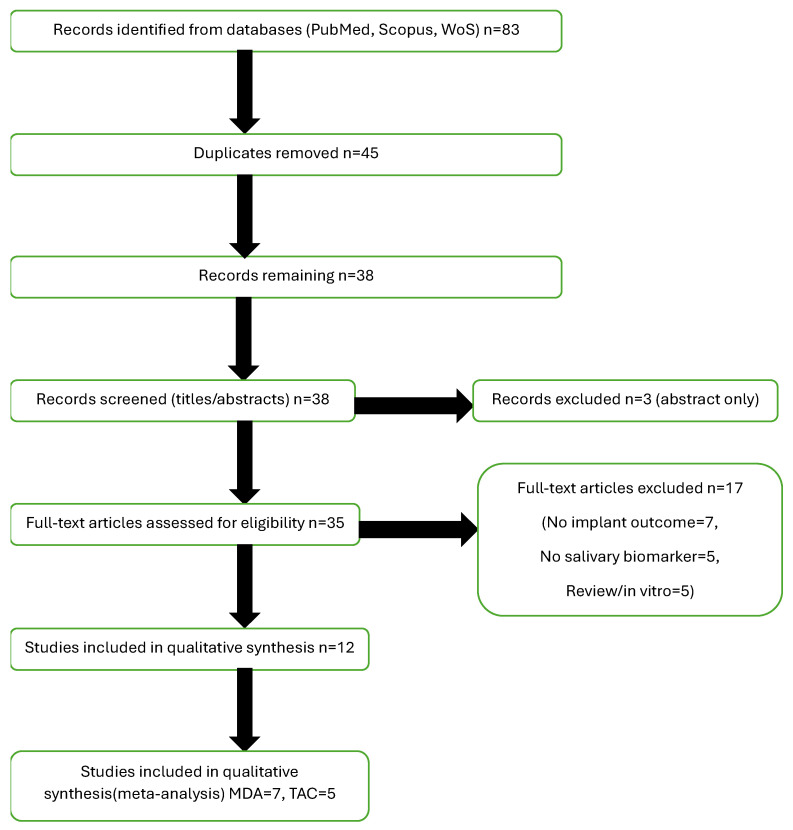
PRISMA Flowchart. Abbreviations: PRISMA, Preferred Reporting Items for Systematic Reviews and Meta-Analyses; MDA, malondialdehyde; TAC, total antioxidant capacity.

**Figure 3 ijms-26-11269-f003:**
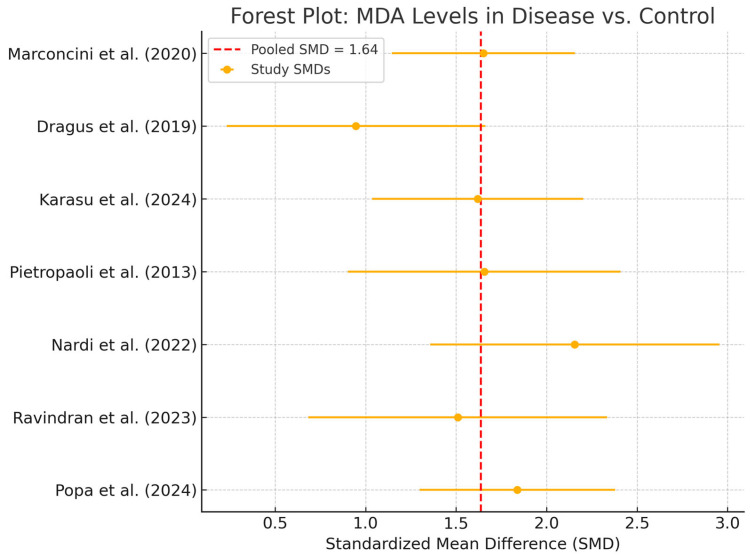
Forest Plot—MDA. Abbreviations: MDA, malondialdehyde; SMD, standardized mean difference [5,13,25,27,28,29,30].

**Figure 4 ijms-26-11269-f004:**
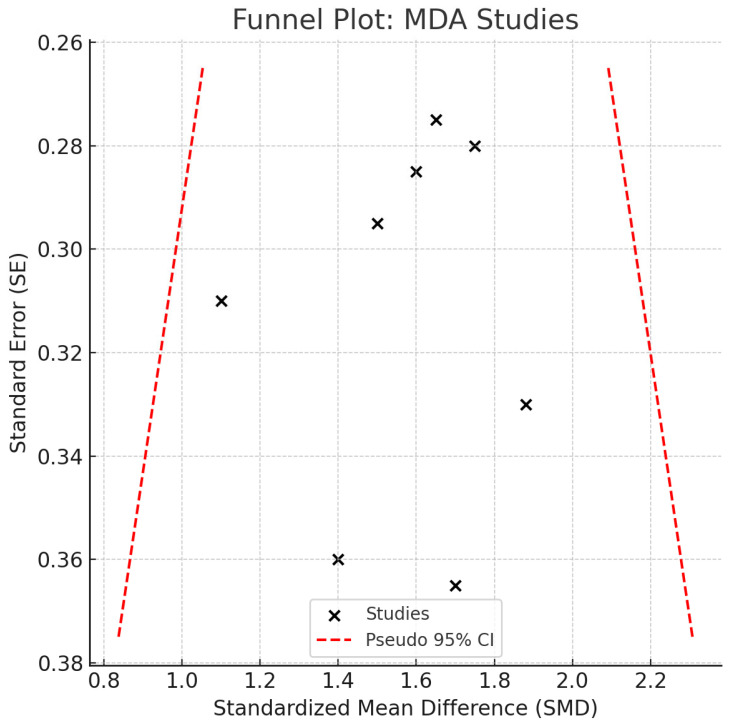
Funnel Plot—MDA. Abbreviations: MDA, malondialdehyde; SE, standard error; CI, confidence interval.

**Figure 5 ijms-26-11269-f005:**
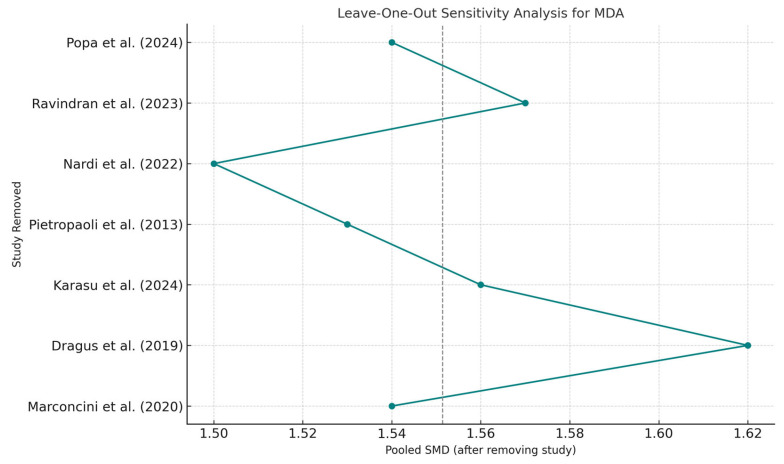
Leave-One-Out Sensitivity Analysis—MDA. Abbreviations: MDA, malondialdehyde; SMD, standardized mean difference [5,13,25,27,28,29,30].

**Figure 6 ijms-26-11269-f006:**
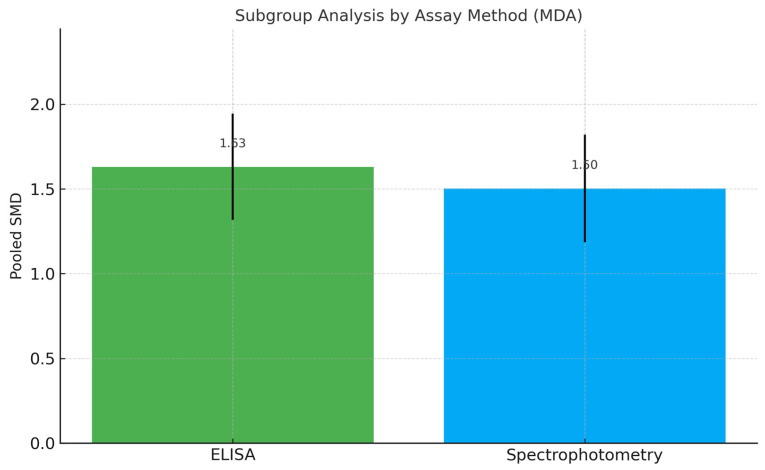
Subgroup Meta-Analysis of MDA Levels by Biomarker Quantification Method. Abbreviations: MDA, malondialdehyde; ELISA, enzyme-linked immunosorbent assay; SMD, standardized mean difference.

**Figure 7 ijms-26-11269-f007:**
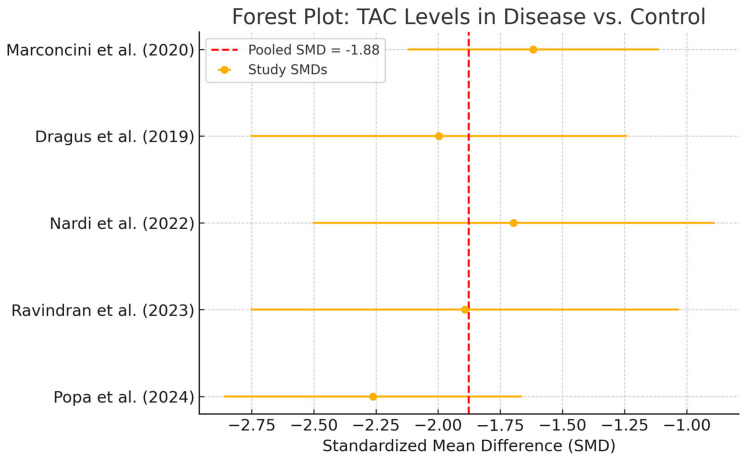
Forest plot—TAC. Abbreviations: TAC, total antioxidant capacity; SMD, standardized mean difference [13,25,27,29,31].

**Figure 8 ijms-26-11269-f008:**
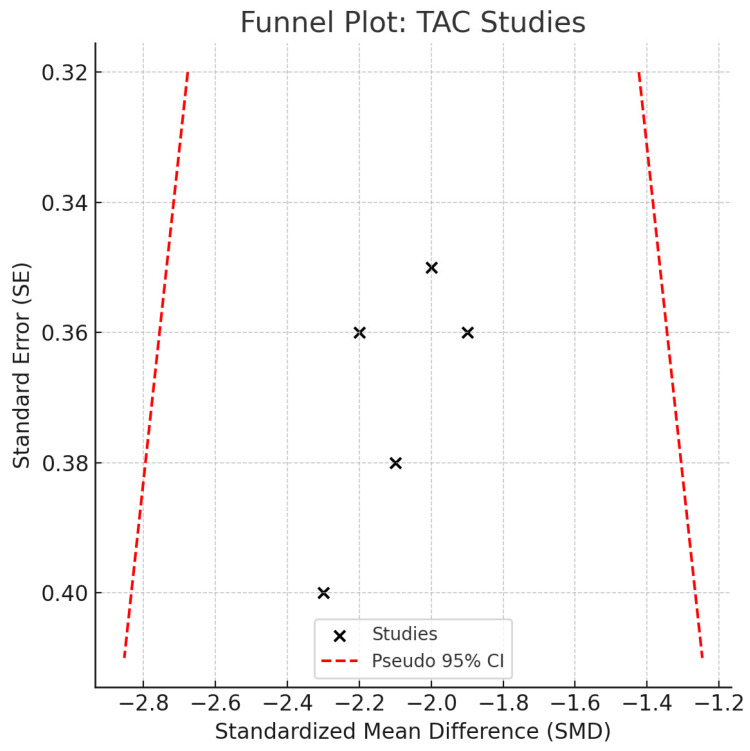
Funnel plot—TAC. Abbreviations: TAC, total antioxidant capacity; SE, standard error; CI, confidence interval.

**Figure 9 ijms-26-11269-f009:**
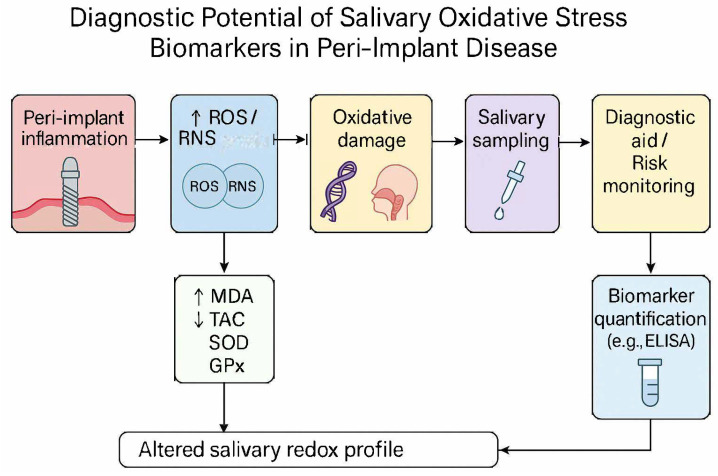
Conceptual framework linking salivary oxidative stress biomarkers with peri-implant disease mechanisms and potential diagnostic integration. This illustration synthesizes current evidence on malondialdehyde (MDA) and total antioxidant capacity (TAC) as complementary indicators of oxidative imbalance. The model is intended for conceptual reference only; clinical thresholds and diagnostic validity require further study. Abbreviations: ROS, reactive oxygen species; RNS, reactive nitrogen species; MDA, malondialdehyde; TAC, total antioxidant capacity; SOD, superoxide dismutase; GPx, glutation peroxidase; ELISA, enzyme-linked immunosorbent assay.

## Data Availability

All data used for synthesis were extracted from publicly available published articles, which are cited in the reference list. Additional details (e.g., search strategy, risk-of-bias assessments, and GRADE evidence profiles) are provided in the Appendix A. Extracted quantitative datasets and pooled effect estimates used in RevMan and MetaXL are available from the corresponding author upon reasonable request.

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
