# Peer review of "Salivary Oxidative Stress Biomarkers in Peri-Implant Disease: A Systematic Review and Meta-Analysis"

_ijms, 2025, doi:10.3390/ijms262311269_

Round 1

Reviewer 1 Report

Comments and Suggestions for Authors

Abstract requires some modifications

Refs of Fig 1???

PICOS model, Refs???

Eligibility Criteria, Refs???

Remove the sub-Sections from Discussion Section

Remove the parts of the text (464-480, and 483-496), as they contain irrelevant data

Refs of Fig. 9???

Conclusion(s) Section: Reduce, and state the main outcomes only....

Reviewer 2 Report

Comments and Suggestions for Authors

The paper ”Salivary Oxidative Stress Biomarkers in Peri-Implant Disease: 2 A Systematic Review and Meta-Analysis 3” by Popa P.S. et al. focuses on evaluating whether salivary malondialdehyde and total antioxidant capacity are associated with peri-implant disease. The manuscript is well organized, and all the criteria for a systematic review have been followed. Nevertheless, there are some issues that I need to address:

  • the quality of the figures included in the manuscript is quite low, and it needs to be improved
  • on lines 61-68 of the introduction section, the authors comment on results obtained in previous studies regarding other oxidative stress biomarkers than the one included in the present article. I think that that part should be removed, as it is not needed.
  • the conclusion section is too long. I recommend summarizing the information.
  • the 12 articles included in the analysis are not cited in the reference section
  • perhaps figures 1 and 9 could be combined, as they contain similar parts
  • minor spelling check should be performed. For example in figure 1 defanse should be corrected and replaces with denfese

Round 2

Reviewer 2 Report

Comments and Suggestions for Authors

In my opinion, the manuscript has been improved by the authors during the revision process, and I agree with its publication.

Author Response

Dear Reviewer,

We thank you very much for your suggestions; they have improved our manuscript considerably.